# Assessing the effectiveness of newborn resuscitation training and skill retention program on neonatal outcomes in Madhesh Province, Nepal

**Mala Chalise** [1], **Ranjan Dhungana** [2‡], **Michael K. Visick** [3‡], **Robert B. Clark** [4] *

**1** Children's Medical Mission, Payson, Utah, United States of America, **2** Safa Sunaulo Nepal, Kathmandu, Nepal, **3** University of Utah School of Medicine, Salt Lake City, Utah, United States of America, **4** Brigham Young University, Provo, UT, United States of America

⊛ These authors contributed equally to this work.
‡ RD and MKV also contributed equally to this work.
\* robclarkmd@outlook.com

**Data Availability Statement:** All data are in the manuscript and supporting information files.

**Funding:** The authors received no specific funding for this work.

## Abstract

Intrapartum events leading to asphyxia at birth are among the leading causes of neonatal morbidity and mortality in Nepal. In response to this, the Nepal Ministry of Health and Population adopted Helping Babies Breathe (HBB) as a tool to improve neonatal resuscitation competencies. The effectiveness of HBB trainings has been well established. However, challenges remain in maintaining skills over time. Safa Sunaulo Nepal (SSN), with support from Latter-day Saint Charities (LDS Charities) designed an initiative for scaling up newborn resuscitation training and skills maintenance over time. This paper reports on the implementation of the SSN model of newborn resuscitation trainings and skill retention, and the changes in perinatal outcomes that occurred after the program. The program built capacity among facility-based trainers for the scale up and maintenance of resuscitation skills in 20 facilities in Madhesh Province, Nepal. A single external Mentor coached and assisted the facility-based trainers, provided general support, and monitored progress. Prospective outcome monitoring tracked changes in health metrics for a period of 14 months. Data was gathered on the neonatal health outcomes of 68,435 vaginal deliveries and 9,253 cesarean sections. Results indicate decreases in neonatal deaths under 24 hours of life (p<0.001), intrapartum stillbirths (p<0.001), and the number of sick newborns transferred from the maternity unit (p<0.001). During the program, facility-based trainers taught resuscitation skills to 231 medical personnel and supported ongoing skill retention. The SSN model for newborn resuscitation training and skills retention is a low-cost, evidence-based program focusing on facility-based trainers who are mentored and supported to scale-up and sustain resuscitation skills over time. Findings from the report are suggestive that the model had a substantial influence on critical neonatal outcomes. Future programs focused on improving neonatal outcomes may benefit by incorporating program elements of SSN model.

**Competing interests:** The authors have declared that no competing interests exist.

## Introduction

The world has made phenomenal progress in reducing child mortality over the past three decades, but global progress was insufficient to universally achieve Millennium Development Goal (MDG) 4. Only one third of MDG countries, which included Nepal, reduced under-five mortality by two-thirds [1].

While worldwide under-five mortality has declined considerably [2], the reductions in neonatal mortality rate have lagged, leading to an increased proportion of overall deaths attributed to the first month of life. Nearly 37% of the estimated child death in 1990 was attributable to neonatal deaths [3], compared with 45% in 2015 [4]. The United Nation (UN) Inter-agency Group for Child Mortality Estimation anticipates that neonatal deaths will account for nearly 52% of child deaths by 2030 [4].

In Nepal, while the under-five mortality decreased from 133 deaths per 1000 live births in 1991 to 39 deaths per 1000 live births in 2016, neonatal mortality only dropped from 50 deaths per 1000 live births in 1991 to 21 in 2016 [5], causing the Neonatal Mortality Rate (NMR) to rise to 61% of the under-five mortality rate [6].

Most neonatal deaths occur within the first week of birth [1]. The first day of life is the most vulnerable time for the baby with more than a quarter (27%) of neonatal death occurring in the first day of life in Nepal [7]. Intrapartum Hypoxic Events (IHE) continue to be one of the leading causes of neonatal morbidity and mortality in low-income countries, including Nepal [8–10]. To achieve Sustainable Development Goal (SDG) 3.2 for reducing the neonatal mortality rate to 12 per 1000 live births, averting preventable deaths is the highest priority.

Strengthening the capacity and competency of healthcare workers to provide timely and appropriate newborn resuscitation is essential to addressing newborn deaths [11]. To this end, the American Academy of Pediatrics (AAP) and other partners developed Helping Babies Breathe (HBB) training to empower healthcare providers with life-saving techniques through simulation-based training [12]. HBB is a cost-effective, simplified neonatal resuscitation curriculum focusing [13] on acquiring and implementing basic resuscitation skills. These skills are sufficient to support the transition of 99% of newborns in low-resource settings, with the help of basic clinical equipment [14]. Multiple studies have documented the effectiveness of HBB in reducing newborn mortality [15–19]. Retention of these skills over time is critical but HBB follow-up studies have shown a decline in knowledge and skills over time [11, 20–22]. Thus, a major challenge in upgrading newborn resuscitation has been maintaining skills over time, as skills training alone is not sufficient. Successful programs have carefully incorporated refresher training, mentoring, and various Quality Improvement (QI) approaches [11, 17, 20, 23–25] to accomplish continuous learning and retention. These have provided guidance on strategies needed to both improve and maintain resuscitation skills.

Following the successful piloting of HBB training in Nepal in 2013 [16], the then Child Health Division of the Ministry of Health and Population (MoHP) incorporated the HBB training curriculum into the in-service training package of Skilled Birth Attendants in 2015. The MoHP has been encouraging the dissemination of improved resuscitation techniques using HBB in other settings as well. Newborn resuscitation using HBB has been supported by multiple child health partners in Nepal. For instance, UNICEF employed HBB to strengthen the resuscitation skills in a portfolio of hospitals during two clinical trials [26, 27]. The National Health Training Center (NHTC), a nodal agency of MoHP for training, is responsible for disseminating approved in-service provider education throughout the country. NHTC has orchestrated HBB Training-of-Trainer (TOT) courses in Nepal since 2012 in collaboration with Latter-day Saint Charities (LDS Charities), the humanitarian arm of The Church of Jesus Christ of Latter-day Saints [28] and Safa Sunaulo Nepal (SSN).

In 2018, SSN, with financial and technical support from LDS Charities, designed a newborn resuscitation training, scale-up, and skill retention program centered on developing and maintaining the capacity of facility-based trainers to sustain the skills required to manage newborn emergencies. The goals of the program were to:

- Utilize evidence-based strategies for scaling up and maintaining resuscitation skills

- Implement a cost-effective program for the scale-up and skill retention

- Monitor the process with key indicators

Compared to other provinces, Madhesh Province is the province in Nepal with the poorest performance in key health indicators including nutrition, teenage pregnancy, family planning coverage, newborn deaths and vaccination coverage. The NMR of the province in 2016 was estimated at 30 per 1000 live births against the national average of 21/1000, with this province accounting for the highest proportion of neonatal deaths in the country [5]. These data emphasized the pressing need for targeted interventions to achieve sustainable progress in key health indicators, including the reduction of neonatal mortality.

Unfortunately, protracted political unrest persisting in the province since 2007, owing to dissatisfaction over province demarcation [29], resulted in a non-conducive environment for training and quality improvement programs to penetrate, thrive, and be sustained as in other provinces. The high number of neonatal deaths in Madhesh, combined with the medically underserved population, made this province a high priority for NHTC, LDS Charities, and SSN to introduce the newborn resuscitation scale-up and retention program. A TOT was organized in 2020 when relative political stability was achieved in the region.

The purpose of this paper is to describe the implementation of the SSN newborn resuscitation training, scale-up, and skill retention program in Madhesh province of Nepal and report changes in newborn outcomes that occurred after the time of program implementation.

## Methods

### Ethics statement

Ethical approval was obtained from the Nepal Health Research Council (Registration number 797/2018) for the newborn resuscitation scale-up and retention program, including data collection from the entire cohort of facilities under the program.

### Study design

We used a prospective cohort design to compare outcomes of birth cohorts in 20 hospitals pre- and post-implementation of the facility-based newborn resuscitation program in the Madhesh province. Data collection spanned a fourteen-month period from October 2020 to November 2021. Inclusion criteria captured facilities that provided delivery services 24 hours a day, 7 days a week and represented a significant proportion of the births in Madhesh province.

To assess the effectiveness of the resuscitation skills scale-up and retention program, the first two months (October- November 2020) and the last two months (October- November 2021) of data collection were considered as baseline period and end-line period, respectively. In the baseline period, newborn resuscitation scale-up and support mechanisms were just beginning and included the HBB TOT, while the end-line period had the scale-up and support mechanisms well established.

**Table 1. Overview of Madhesh Province.**

| Overview of Madhesh Province | |
|---|---|
| Total number of districts | 8[a] |
| Total Population | 5,404,145[a] |
| Total number of public hospitals (Secondary and Tertiary Hospital) | 13[a] |
| Neonatal Mortality Rate | 30 per 1000 live births [b] |

[a]Data from Madhesh Province Profile [30]

[b]Data from Nepal Demographic Health Survey 2016 [5]

Madhesh province lies in the southern plain of the country. The province is less topographically challenged in comparison to other provinces that have complex terrain. However, the province lags far behind other provinces in terms of key health indicators due to the concerns previously mentioned.

## Study population and setting

The newborn resuscitation training and retention program was conducted in Madhesh province, one of the seven provinces under the new federal structure in Nepal. An overview of the province is provided in Table 1.

## Description of intervention

The newborn resuscitation training and skill retention program was implemented in three phases (Fig 1). Details of each phase are presented below:

**TOT phase.** The TOT phase focused on developing newborn resuscitation competencies of facility-based trainers with the first TOT held on 27th- 30th November and the second TOT course from 2-5th December 2020. NTHC selected the facilities and trainers in coordination with the Provincial Health Directorate. The HBB TOT was facilitated by 10 Nepali trainers who used a HBB and Bleeding After Birth Complete (BABC) (BABC is a module of the Helping Mothers Survive training curriculum developed by JHPIEGO. The module is designed to reduce maternal deaths due to the most common cause of maternal intrapartum death, postpartum hemorrhage (PPH). The module builds the skills of healthcare providers working in clinical settings, including communication and teamwork, active management of the third stage of the labor, early detection and basic management of PPH, and advanced PPH care skills.) Learning Resource Package, developed by NHTC and SSN, as a field trial.

A mix of public and private health institutions with large delivery services in Madhesh province participated in the program to improve and sustain neonatal resuscitation skills. Eighty eight facility-based trainer, candidates, representing 37 facilities, attended. Seventy-one out of the 88 (81%) were nurses by profession with the majority (45%) of the candidates working in their profession for one to five years. Forty-three out of 84 participants (51%) were from Secondary Hospitals, with 4 participants not completing the registration survey. Forty-six out of 84 (55%) indicated they personally attended up to 25 deliveries in an average month. Seventy-two out of 84 (86%) mentioned they resuscitated babies with a bag and mask as part of their current job.

Validated, standardized tools were used to conduct pre- and post-knowledge tests of the trainers. The knowledge test had 18 questions with a passing score of 15. The average knowledge pre-test score was 16 while the average post-test score was 17 with 6% change in pre- to post-knowledge test. Eighty two percent of participants had a passing score or higher in pretest with 94% passing the post-test. SSN and LDS Charities provided the new facility-based trainers with hard and soft copies of instructional materials such as flipcharts, provider guidebooks, and testing materials. Training equipment consisted of NeoNatalie manikins, Penguin

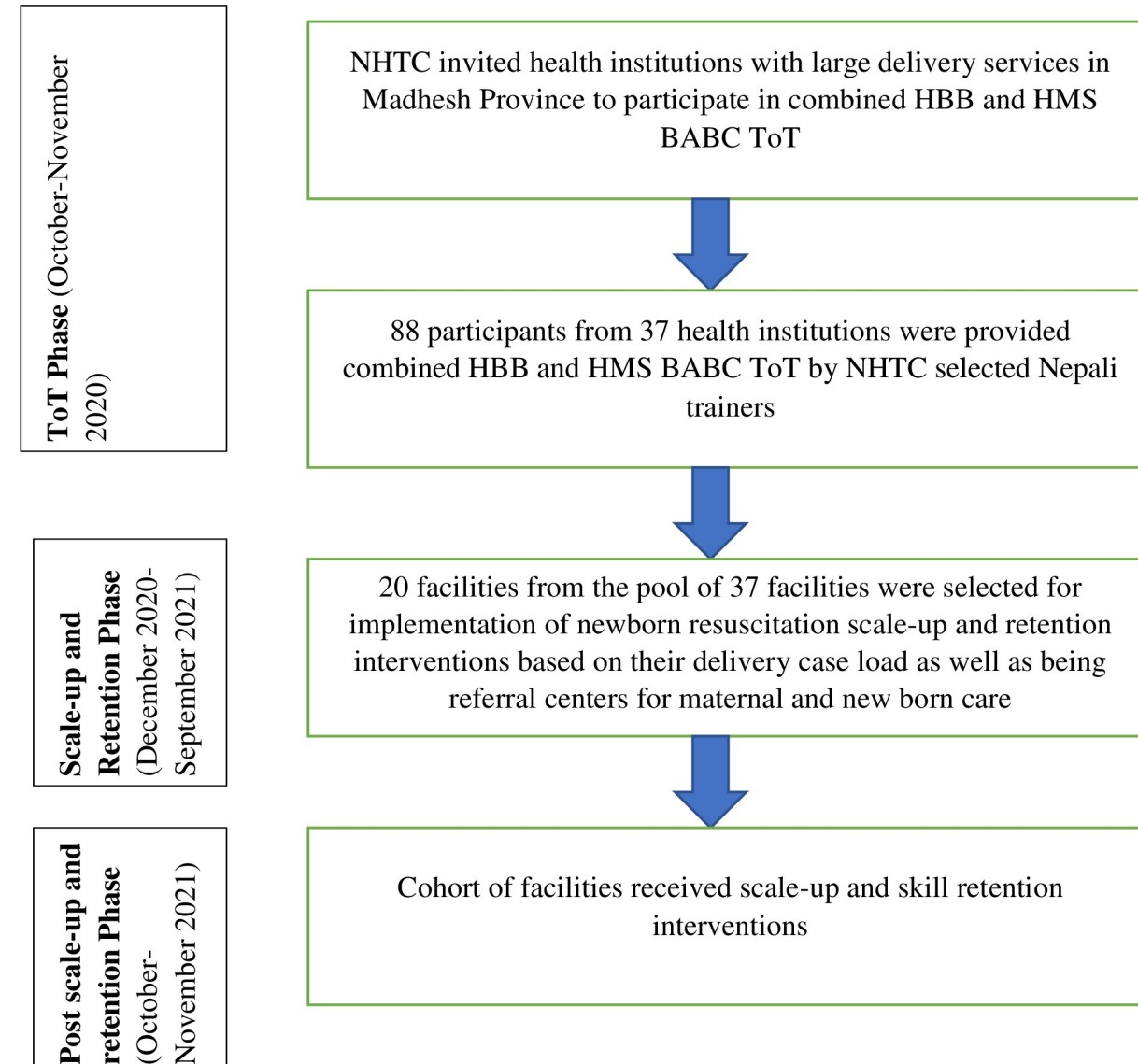

**Fig 1. Newborn resuscitation training and skill retention program algorithm.**

suction devices, bag and masks, and stethoscopes. Clinical equipment, such as bag and masks, suction devices, and stethoscopes for the delivery rooms were also supplied.

The facility-based trainer candidates were empowered to facilitate scale-up of resuscitation training to all physicians and nurses/midwives in their facilities and were expected to create an enabling environment for retention of learned skills.

**Scale-up and skill retention phase.** Following the TOT, 20 of the 37 facilities were selected for the active assistance with scale up and retention, based primarily on high delivery volume. An experienced HBB trainer with research experience was recruited to serve as a mentor, teacher, monitor of facility-based indicators, and administrative liaison for the cohort of 20 facilities (external Mentor). The scale-up and skill retention phase focused on facility-based trainers cascading resuscitation skills to staff in all 20 facilities, and then working to retain the learned skills, with the help of the equipment and materials provided by SSN and LDS Charities during the training of trainers.

In the cohort of 20 facilities, the external Mentor assisted with facility-level scale-up of resuscitation training to all physicians and nurses/midwives. The Mentor also assisted with a) practice sessions, review meetings, and refresher training b) on-site coaching c) monitoring key indicators and d) liaison with administration. The Mentor was the driver for scaling up newborn resuscitation in the cohort of facilities through utilization of evidence-based methods to develop staff capacity.

Vaginal delivery (normal and assisted) was provided by all 20 facilities in the mentoring cohort, and 17 provided cesarean section delivery. The labor units in the hospitals were led by physicians and midwives and equipped to provide neonatal resuscitation at birth. Pediatricians led neonatal care units at Tertiary Hospitals, while in primary and Secondary Hospitals, sick newborns were managed in the pediatric unit, which was led by either medical officers or nurses/midwives.

The facility-based trainers, under the supervision of the Mentor, conducted facility-level HBB trainings to physicians and nurses/midwives in the labor ward and neonatal care unit. Training also bridged newborn resuscitation knowledge and skill gaps among new staff due to staff turnover. The facility-based trainers encouraged daily practice sessions using low-dose high-frequency (LDHF) bag and mask ventilation practice in their respective facilities, recorded the practice performance, established of functional practice corners, and supervised regular staff practice and refresher training. Other interventions used by facility-based trainers for skill retention included weekly review meetings where resuscitation debriefing was done.

The HBB Mentor reviewed the practice logs and conducted Objective Structured Clinical Examinations (OSCEs) (OSCEs are formal clinical performance assessments that evaluate the learner's clinical competence using simulated scenarios in which the learners' skills are rated by an observer while they demonstrate their skills. OSCEs and the Bag and Mask Skill Checks are standard assessment tools in the HBB curriculum) and Bag and Mask Skill Check drills during a monthly monitoring visit. The Mentor also conducted a weekly telephone meeting with facility-based trainers and monthly in-person meetings with facility-based trainers, the nursing in-charge, and the medical superintendent to review the progress of HBB training and conduct further planning. Other activities included assessment of required support such as practice and clinical equipment and liaising with hospital administration to create a conducive environment for skill retention. The Mentor was also responsible for obtaining monthly program data from the cohort of facilities.

Quality improvement was jointly monitored by the facility-based trainers, the Mentor, and the hospital nursing supervisor. Together they observed the HBB skills of staff using a neonatal simulator during the Mentor's regular and unannounced site visits and reviewed the data from the practice logbook, on-site coaching, delivery and newborn logbooks, and refresher training.

## Post scale-up and retention phase

During the post-intervention phase, the facility-based trainers continued with interventions from the scale-up and retention phase. However, the Mentor did not support facility-based trainers in doing so. The Mentor obtained monthly program data during this phase.

## Data management

**Outcomes.**   The primary outcomes included the pre-post difference in a) the intrapartum stillbirths and b) neonatal deaths within the first 24 hours of life. Stillbirths were considered fresh if they were not macerated. Secondary outcomes included differences in neonatal deaths after first 24 hours of life, the utilization of bag and mask ventilation, the number of sick newborns transferred from the maternity unit, and the percentage of staff trained with HBB.

**Data collection.** Monthly facility-level metrics included the number of vaginal deliveries, the number of caesarean sections, the number of intrapartum (fresh) and macerated stillbirths, the number of babies not breathing at birth, and the number of neonatal deaths less than and more than 24 hours of life. These were MoHP mandated metrics. In the tertiary level facilities, monthly facility-level data were derived from the health information department, which tabulated and reported MoHP-mandated metrics. In smaller facilities, the mentor gathered data from registers or logs maintained in the labor room and in the maternity ward. The data thus gathered was certified by the concerned health institution to ensure data quality.

Additional facility level metrics that were not MoHP mandated included the number of sick newborns transferred from the maternity unit and the number of babies receiving assisted ventilation with bag and mask (B&M). Sick newborns transferred from the maternity unit were a proxy for all-cause newborn morbidity during the first days of life. These data were gathered from registers or logs maintained in the labor room and in the maternity ward.

The hospitals in the cohort had no prior system of recording the number of babies receiving B&M ventilation for the pre-intervention period, although the hospitals reported practicing B&M ventilation. As these hospitals started recording this metric only after the HBB TOT, the sum of the measure from the first two months of the scale-up and retention phase (December 2020 and January 2021) was compared with the sum of measures from the post-intervention period for this metric only. Personal data of patients, such as demographic information, were not collected.

Monthly monitoring reports generated by the Mentor documented the scale-up, facility support, and facility-level metrics, including the number of providers trained/retrained, maintenance of an equipped practice corner, low-dose high-frequency (LDHF) practice logs, completion of self-evaluation forms, review meetings held, and review of delivery and newborn logs. Since these metrics were not mandated by MoHP, the mentor collected these data on a monthly basis from the facility-based trainers, who were responsible for record keeping.

**Data analysis.** Facility-level metrics were analyzed by comparing the baseline and endline measurements (S1 Data). The baseline measurements were the sum of data for the initial two months (TOT phase) and the endline measurements were the sum of data from the final two months (post scale-up and skill retention phase). A Paired T-Test was conducted to determine if baseline and endline values were significantly different. The number of deliveries and C-sections were analyzed as potential confounding factors.

Interhospital differences in perinatal outcomes were also analyzed by the type of the hospital. After the Mentor collected data into monthly reports, the Mentor conducted data quality and integrity checks. The monthly reports were forwarded to SSN and to LDS Charities with SSN conducting an initial review, analysis, and preparation of mandated reports. A Nepal-based research assistant combined the data into a master file, clarified missing variables with the Mentor, deleted variables missing significant data, changed text into appropriate numerical values, and conducted data analysis using SPSS version 21.

## Results

### Facility profile

The 20 facilities in the newborn resuscitation and scale-up program cohort represented a mix of public and private hospitals. The Government of Nepal has categorized public hospitals into different categories according to the level of government [31]. Accordingly, the facilities in the study cohort consisted of four different facility types: Primary Hospital (5), Secondary Hospital (10), Tertiary Hospital (3) and Private Hospital (2).

**Table 2. Outcomes by facility type.**

| Facility Type | Primary Hospital | | Secondary Hospital | | Tertiary Hospital | | Private Hospital | | Total |
|---|---|---|---|---|---|---|---|---|---|
| Outcomes | N | % | N | % | N | % | N | % | |
| Deliveries (except C-section) [a] | 8818 | 12.89 | 21198 | 30.98 | 35269 | 51.54 | 3150 | 4.60 | 68435 |
| C-section | 764 | 8.26 | 2403 | 25.97 | 5454 | 58.94 | 632 | 6.83 | 9253 |
| Babies not breathing at birth | 1071 | 16.01 | 2401 | 35.90 | 2790 | 41.72 | 426 | 6.37 | 6688 |
| Babies receiving Bag and Mask Ventilation | 632 | 16.51 | 1273 | 33.25 | 1701 | 44.44 | 222 | 5.80 | 3828 |
| Number of intrapartum stillbirths | 92 | 10.69 | 194 | 22.53 | 535 | 62.14 | 40 | 4.65 | 861 |
| Neonatal deaths within 24 hours | 54 | 11.76 | 134 | 29.19 | 246 | 53.59 | 25 | 5.45 | 459 |
| Neonatal deaths after 24 hours | 18 | 10.17 | 45 | 25.42 | 105 | 59.32 | 9 | 5.08 | 177 |
| Sick newborns transferred from the maternity unit | 1026 | 17.88 | 2040 | 35.56 | 2323 | 40.49 | 348 | 6.07 | 5737 |

[a] Cesarean section

According to the Health Infrastructure Development Standards 2017, Primary Hospitals in Nepal are mandated to provide normal delivery and comprehensive emergency obstetric and neonatal care (CEONC) [31]. However, in practice, availability of CEONC services in the Primary Hospital depends upon the availability of physicians. Only two out of the five Primary Hospitals in the study cohort provided CEONC services. Besides providing normal delivery and CEONC services, both Secondary and Tertiary Hospitals have dedicated departments and wards for providing gynecological and obstetric services. Unlike Secondary Hospitals, Tertiary Hospitals have advanced infrastructures and human resources for providing maternal and newborn care, including a higher number of beds, neonatal care units, and pediatricians. Delivery services are supervised by obstetrician-gynecologists in both Secondary and Tertiary Hospitals.

A total of 68,435 vaginal deliveries and 9,253 Cesarean deliveries took place at the 20 health facilities during the study period. Tertiary Hospitals accounted for more than half of the vaginal deliveries (51.54%), C-section deliveries (58.94%), neonatal deaths within 24 hours of life (53.59%), neonatal deaths after 24 hours of life (59.32%) and intrapartum stillbirths (62.14%). Sick newborns transferred from the maternity unit (40.49%), babies not breathing at birth (41.72%) and babies receiving bag and mask (44.44%) were also higher in Tertiary Hospitals as compared to other facility types. Outcomes according to facility type are listed in Table 2.

### Effect of newborn resuscitation scale-up and skill retention program on perinatal outcomes

To test the hypothesis that baseline and endline means for perinatal outcomes were equal, a dependent samples t-test was done. Before preforming the analysis, the assumption of normally distributed difference scores was done to examine whether the assumption was satisfied. Data for each perinatal outcome satisfied the assumption of normally distributed pre-post difference scores as the skewness was $<|2.0|$ and kurtosis $<|9.0|$ [32].

Since correlation between baseline and endline measures were significant at p-value $< .001$ (Table 3), the dependent sample t-test was considered an appropriate statistical test for all perinatal outcomes.

The null hypothesis of equal baseline and endline means was rejected for neonatal deaths $<24$ hours $t(19) = 6.212$, $P<0.001$, intrapartum stillbirths $t(19) = 5.225$, $p<0.001$, and sick newborns transferred from the maternity/pediatric unit $t(19) = 5.358$, $p<0.001$. Thus, the endline means for neonatal deaths $<24$ hours, intrapartum stillbirths and sick newborn transferred from the maternity unit were statistically lower than the baseline means suggesting the potential impact of newborn resuscitation and scale-up program in the observed association.

**Table 3. T-test and suitability of dependent sample t-test.**

| Perinatal outcomes | Baseline Mean (n = 20) | Standard deviation | Endline Mean (n = 20) | Standard deviation | correlation | t-score |
|---|---|---|---|---|---|---|
| **Babies not breathing at birth** | 46.80 | 48.22 | 38.40 | 33.37 | 0.877 (p<0.001) | 1.514(19) (p = 0.146) |
| **Babies receiving Bag and Mask Ventilation** | 32.55 | 39.23 | 30.15 | 27.79 | 0.853 (p<0.001) | 0.505 (19) (p = 0.619) |
| **Intrapartum stillbirths** | 10 | 14.07 | 4.3 | 9.58 | 0.987 (p<0.001) | 5.225(19) (p<0.001)* |
| **Neonatal deaths within 24 hours** | 5 | 5.73 | 2.2 | 4.71 | 0.966 (p<0.001) | 6.212 (19) (p<0.001)* |
| **Neonatal deaths after 24 hours** | 1.7 | 2.68 | 1.05 | 1.79 | 0.848 (p<0.001) | 1.942 (19) (p = 0.067) |
| **Sick newborns transferred from the maternity unit** | 54.65 | 41.45 | 36.45 | 30.71 | 0.955 (p<0.001) | 5.358 (19) (p<0.001)* |

* statistically significant observation

The first day neonatal mortality showed a declining trend, (Fig 2) with a decrease of approximately 56% (p<0.001) neonatal deaths. Overall, there were 100 neonatal deaths during the baseline period, compared to 44 in the endline period.

During the study period, overall intrapartum stillbirths decreased by 57% (p<0.001), from 200 during the baseline period to 86 in the follow up period (Fig 3).

Sick newborns transferred from the maternity unit included their transfer to another unit of the hospital or to another hospital or discharged home. Sick infants transferred from the maternity unit was a proxy for all-cause newborn morbidity during the first days of life. Overall, sick newborns transferred from the maternity unit decreased by approximately 33% over time from 1093 in the baseline period to 729 in the endline period (p<0.001) (Fig 4).

Inter-hospital difference in perinatal outcomes was observed for neonatal deaths within 24 hours, intrapartum stillbirths and sick newborns transferred from the maternity unit (Table 4).

Neonatal death within 24 hours of life was found to be significantly lower in Primary Hospitals (p = 0.0577) and Secondary Hospitals (p = 0.0239) during the endline period as compared to the baseline period. The decline was approximately 73% (p = 0.0577) in Primary Hospitals and nearly 64% (p = 0.0239) in Secondary Hospitals. The intrapartum stillbirths were observed to be significantly lower in Primary Hospitals (p = 0.0276) and Private Hospitals (p = 0.0136). There was an approximately 79% (p = 0.0276) decrease in the number of intrapartum stillbirths during endline period (n = 29) as compared to the baseline period (n = 6) in the five Primary Hospitals. The number of intrapartum stillbirths in the two Private Hospitals decreased by approximately 92% (p = 0.0313) from 13 in baseline to 1 in the endline period. Similarly, sick newborns transferred from the maternity unit decreased by nearly 27% (p = 0.0060) over time in Tertiary Hospitals.

## HBB scale-up and support

The monthly HBB scale-up and support measures were largely comprised of qualitative indicators. The scale-up of HBB resulted in 231 providers receiving the training during active mentoring. The facility-based trainers, with the assistance of the Mentor, cascaded the training in their facilities by training existing and new staff during the program, and holding refresher training sessions.

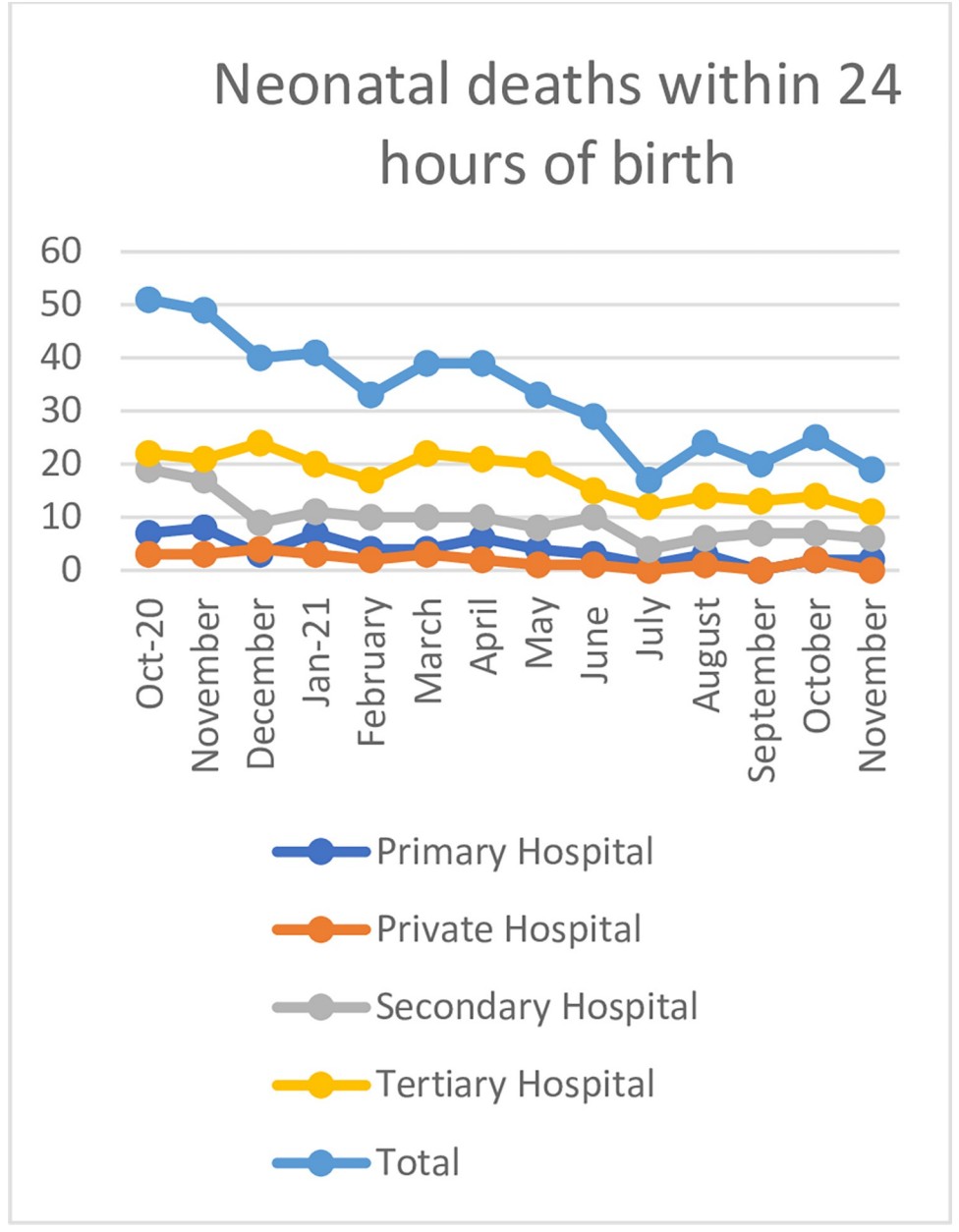

**Fig 2. The run chart depicting the association of HBB program with neonatal mortality under 24 hours birth over the program period.**

At the end of the reporting period, all facilities reported fully functional practice corners that were equipped and utilized, were holding periodic review meetings, conducting self-evaluation surveys, and maintaining both delivery and newborn logs.

## Discussion

SSN, with support from LDS Charities, set up a program to scale-up and maintain newborn resuscitation skills in Madhesh province of Nepal. This paper reports on the initiative, compares the newborn metrics and HBB scale-up activities from a cohort of facilities from

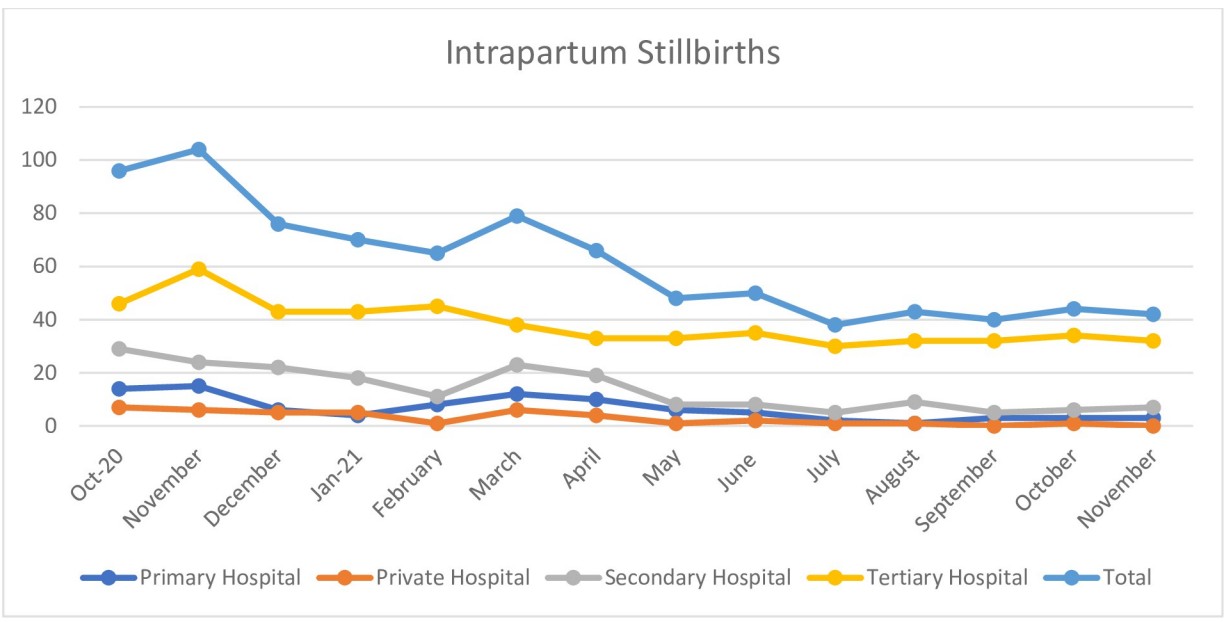

**Fig 3. The run chart depicting the association of HBB program with intrapartum still birth over the program period.**

Madhesh province, and analyzes the changes in outcomes associated with improving the quality of newborn resuscitation at the facility level.

Analysis of baseline and endline data from this cohort of facilities suggests that the SSN program for the scale up and retention of newborn resuscitation skills was associated with marked decreases in the number of neonatal deaths within 24 hours, intrapartum stillbirths, and the number of sick newborns transferred from the maternity unit. Findings of this initiative

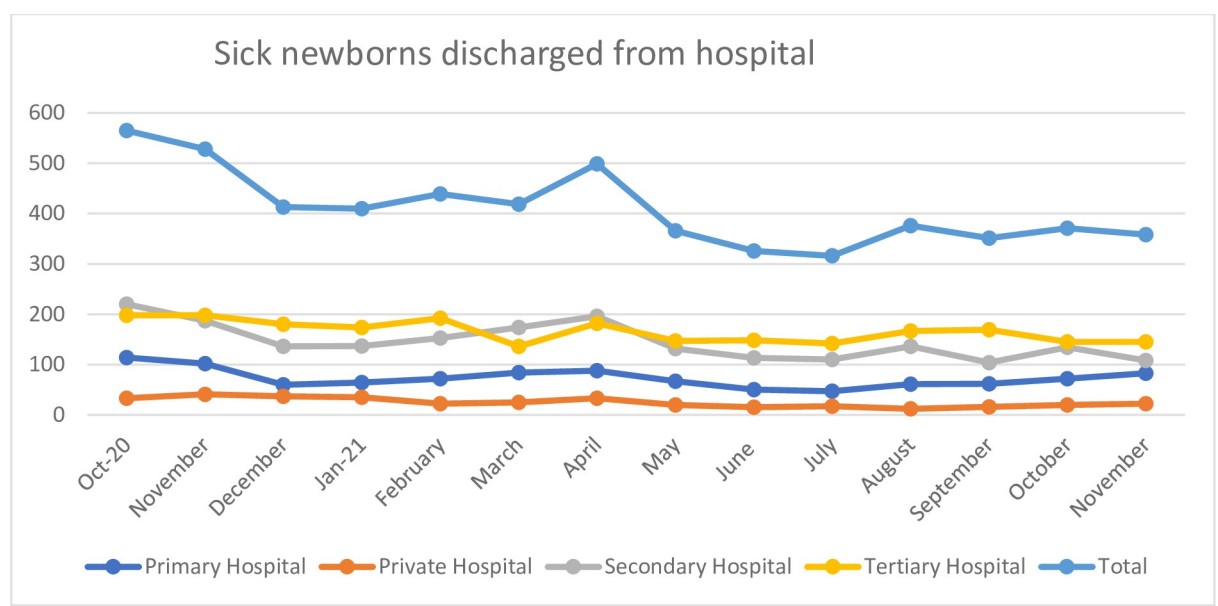

**Fig 4. The run chart depicting the association of HBB program with sick newborns transferred from the maternity/pediatric unit over the program period.**

**Table 4. T- test scores of perinatal outcomes by hospital.**

| Facility Type | Primary Hospital | | | Secondary Hospital | | | Tertiary Hospital | | | Private Hospital | | |
|---|---|---|---|---|---|---|---|---|---|---|---|---|
| Outcomes | Baseline Mean | Endline Mean | p-value | Baseline Mean | Endline Mean | p-value | Baseline Mean | Endline Mean | p-value | Baseline Mean | Endline Mean | p-value |
| Babies not breathing at birth | 71 | 75.5 | 0.581 | 168.5 | 125 | 0.242 | 203 | 155.5 | 0.242 | 25.5 | 28 | 0.5 |
| Babies receiving Bag and Mask Ventilation | 45 | 62.5 | 0.090 | 100 | 96 | 0.295 | 154 | 122.5 | 0.092 | 26.5 | 20.5 | 0.128 |
| Number of intrapartum stillbirths | 14.5 | 3 | 0.027 | 26.5 | 6.5 | 0.0069 | 8.5 | 7 | 0.349 | 6.5 | 0.5 | 0.013 |
| Neonatal deaths within 24 hours | 7.5 | 2 | 0.058 | 18 | 6.5 | 0.023 | 21.5 | 12.5 | 0.079 | 3 | 1 | 0.295 |
| Neonatal deaths after 24 hours | 2.5 | 1 | 0.204 | 5 | 2 | 0.204 | 8.5 | 7 | 0.349 | 1 | 0.5 | 0.711 |
| Sick newborns transferred from the maternity unit | 108 | 77.5 | 0.065 | 103.5 | 121 | 0.868 | 198 | 145.5 | 0.006 | 37 | 21 | 0.138 |

correspond with several previous studies that have noted significant declines in intrapartum stillbirth [16, 17, 33–36] and neonatal mortality within 24 hours of life [16, 17, 33, 37, 38], in part due to the focus in the HBB curriculum on identification and resuscitation of seemingly lifeless neonates at birth.

As observed in other studies, neonatal deaths after 24 hours did not change significantly after the implementation scale-up and skill retention [15]. A plausible explanation could be that HBB focuses more on interventions to restore spontaneous breathing right after birth than on interventions to enhance postnatal survival. Appropriate ventilatory support will assist the apneic infant to commence breathing and support the breathing of infants who are not breathing well. Unfortunately, it does not completely reverse the effects of severe asphyxia, or the pathology of pulmonary prematurity, infections, malformations, or other newborn complications. Therefore, in severely compromised infants, resuscitation may lead to survival of asphyxiated babies during the first day of life only. The focus of the HBB curriculum on preventing hypothermia and preventing infection may lead to increased survival after the first 24 hours of life that was not detected in this study due to poor availability of data on babies after their transfer to advanced care. Additional interventions that address the other causes of mortality in the neonatal period are necessary to reduce neonatal mortality after 24 hours from birth [15].

Analysis of inter-hospital differences in perinatal outcomes suggest that resuscitation scale-up and retention had greater impact in Primary Hospitals in terms of reduction in 24-hour neonatal death and intrapartum stillbirths, as compared to other hospitals. Secondary, Tertiary and Private Hospitals may have more advanced technologies for newborn care and/or more specialized providers than Primary Hospitals. While the care in Secondary and Tertiary Hospitals is primarily provided by nurses/midwives, they may also have greater experience in newborn resuscitation due to frequent exposure to compromised infants based on both delivery volumes and referral center status. In Primary Hospitals, newborn care is often provided in relatively resource-constrained facilities by lower cadre health professionals, mostly by nurses, who may not have a foundation of knowledge and skills in newborn resuscitation. Hence, the greater effectiveness of scale-up and skill retention in Primary Hospitals may be a reasonable expectation. A study that compares perinatal outcomes in a first-level referral hospital before and after the implementation of HBB training has also shown that HBB training is effective in a local first-level referral hospital with few resources, such as Primary Hospitals in the context of Nepal, in improving key perinatal outcomes [34]. Similarly, Private Hospitals, where

training opportunities are generally lower as compared to Public Hospitals, showed considerable improvement in reduction of intrapartum stillbirths with HBB training, suggesting substantial potential impact for the newborn resuscitation training and skill retention program in Private Hospitals in Nepal.

Notably, 86% of facility-based trainers in the TOTs were already resuscitating babies in their current job at the time of training. Further, the pre-test scores demonstrated a good understanding of newborn compromise and resuscitation. Nevertheless, neonatal outcomes improved significantly in spite of the prior knowledge and experience. This emphasizes that training alone is insufficient to achieve gains in neonatal outcomes and needs to be followed by regular practice, mentorship, retraining, and monitoring—the main focus of our efforts.

Run charts tracking perinatal outcomes over time showed considerable improvements towards the end of the post scale-up and skill-retention phase, indicating the possibility of continued improvements even after the external support for newborn resuscitation scale-up and skill retention ended. However, it is difficult to establish the long-term sustainability of the newborn resuscitation program from this program, as the post scale-up and retention phase follow up was conducted for only two months. Towards the end of the program, the Mentor focused on transitioning to internal mentoring efforts through selection of one facility-based trainer per hospital as a focal person to continue the institutionalization of skill-retention and activities such as ensuring functionality of the practice corner, training new staff, encouraging daily practice sessions, and conducting refresher trainings. All 20 hospitals in the study cohort had a focal person appointed by the end of the program. Success of the internal mentoring efforts in achieving sustainability of the newborn resuscitation scale-up and skill retention program would also depend on continuous support from hospital clinical leadership.

Newborn resuscitation training using HBB was first launched in Nepal in early 2012 under the sponsorship of LDS Charities. The program introduced the HBB curriculum to physicians, nurses, and hospital leaders under the MoHP [39]. Since then, HBB TOTs have been held in all seven provinces of the country with various methods of skill retention [40].

A deterioration over time in the retention and the application of new life-saving skills by providers has been a major challenge to sustainability, mainly due to lack of motivation by, and lack of support for TOT graduates to scale-up HBB after initial training. We focused on assisting facility-based trainers to take the lead in gradually scaling up and sustaining the skills through on-the-job cascade trainings to all service providers, encouraging regular practice sessions, and refresher trainings.

The external Mentor used evidence-based strategies for supporting and scaling up HBB training. These included monthly monitoring visits to review practice logs; conducting OSCEs and Bag and Mask Skill Checks; supportive supervision; weekly telephone meetings with facility-based trainers; and monthly in-person meetings with hospital clinical leadership to facilitate HBB skill retention by providers through necessary equipment and training support.

To our knowledge, the interventions rolled out under this program for skills retention are unique to the country and offer numerous advantages to resource-limited settings like Nepal. First, the SSN and LDS Charities model for a newborn resuscitation scale-up program enabled improved assisted ventilation by encouraging hands-on low-dose high- frequency practice with a simulator for retention of learned skills. The model focused on the role of an external Mentor assisting the facility-based trainers in cascading HBB training to all providers, and holding practice sessions, on-site coaching, and refresher trainings. Evidence from other low and middle-income countries has also demonstrated that HBB training with frequent refresher training and on-the-job practice can positively impact the retention of knowledge and skills for newborn resuscitation and yield remarkable improvement in neonatal mortality outcomes [41].

The model also emphasized the role of the Mentor in collaboration with local clinical leadership to establish a system for ongoing mentorship and supervision, and to foster policies that support institutionalization of HBB training. The crucial role of a dedicated Mentor in quality-improvement and scale-up has been acknowledged in the scale-up of various maternal and child health interventions [42].

The SSN and LDS Charities model of newborn resuscitation scale-up proved to be a low-cost intervention. The cascade training approach by facility-level providers incurred much lower direct cost to SSN and LDS Charities by avoiding travel costs and Daily Subsistence Allowance (DSA) expenses. Similarly, indirect costs—such as loss of work hours by the hospital as part of sending providers for off-site training—was also remarkably reduced as the facility-level providers were trained while they were still on-the job. Further, the expense of the single, full-time Mentor was diffused among 20 facilities.

Finally, a strength of the program was its reach from tertiary level hospitals to peripheral level and private health institutions with high volumes of deliveries in the province. These volumes constituted a huge proportion of the deliveries in the province, thereby influencing the clinical practice of newborn resuscitation at large scale.

The resuscitation scale-up and skill retention efforts faced challenges due to the COVID-19 pandemic. The program took place when Nepal was hard hit by the global COVID-19 crisis. All the Tertiary Hospitals in the study cohort were designated as COVID-19 hospitals for the province. Similarly, Primary and Secondary Hospitals also faced the challenges of managing the cases of COVID-19 and providing essential health care services without concomitant increases in staffing. As part of the structural adjustment to the challenges posed by COVID-19, the facility-based trainers as well as providers in the study cohort underwent task sharing and restructuring to other units of the hospital. As a result, cascade training to facility-based providers occurred very sporadically, if at all, during three months of the scale-up and retention phase, from December 2020 to February 2021. Hence, the number of facility-level providers trained in the study period was lower than in other similar programs [43]. Yet, improvement in major perinatal outcomes such as neonatal deaths within 24 hours, intrapartum stillbirths and number of sick newborns transferred was observed despite irregular practice, mentorship, and cascade training during those three months.

Another challenge related to the data collection process itself. The program monitored progress by using limited indicators, most of which were tracked by the facilities for routine reporting to the MoHP and therefore less prone to error—neonatal deaths being one of those metrics. However, even though most of the facilities in the cohort tracked stillbirths, misclassification and underreporting of fresh stillbirths was widespread. Accordingly, delivery logs had to be reviewed to confirm accuracy in the classification of fresh stillbirths. Similarly, since the facilities did not track and report on resuscitation data for the MoHP, delivery logs had to be reviewed to obtain such data. Sick newborns transferred from the maternity unit—a proxy indicator for all-cause morbidity—were also not tracked by facilities as part of routine reporting to the MoHP. This necessitated a review of newborn logs to determine the number of sick infants transferred.

Due to the design as a skill training and retention program, rather than a clinical study, there was no control/comparison group, which restricted the ability to establish the causality of the program on observed perinatal outcomes. This report describes the changes associated with newborn resuscitation scale-up on perinatal outcomes using routinely collected data, without establishing the causality, as in clinical trials. With this, we are unable to confirm whether the observed outcomes were solely attributable to the newborn resuscitation scale-up and skill retention program or to other changes at the facilities during the time period. Another limitation of the study relates to the use of routinely collected data. Our study utilized

routinely collected facility-level metrics, obtained for the newborn resuscitation training, scale-up and skill retention program monitoring without a prior specific research goal. As such, data were not available for key maternal and fetal factors that may have significantly affected the findings in the study. Therefore, the study is unable to ascertain the role of maternal and fetal factors in the observed association. Hence, the findings from this report should be interpreted and generalized with caution.

This report highlights that the SSN model of newborn resuscitation scale-up and skill retention could be a potential evidence-based, cost-effective model for improving perinatal outcomes in low-resource settings through building and sustaining in-facility capacity for resuscitation.

## Conclusions

This study shows that implementation of the SSN program for newborn resuscitation scale-up and skill retention is associated with reductions in neonatal deaths within 24 hours, intrapartum stillbirths, and sick newborns transferred from the maternity unit, and improved clinical practices in Madhesh province. The analysis confirmed that this approach to newborn resuscitation skill retention and scale-up is a simple, low-cost program focusing on facility-based cascade training in newborn resuscitation skills for providers using the HBB curriculum, LDHF practice, and subsequent mentorship. The program also demonstrated the role of mentorship in capacity building, liaison with clinical leaders, scale-up, monitoring, and quality enhancement with a single Mentor supervising and monitoring 20 facilities. To conclude, this model has the potential to significantly improve perinatal outcomes and could guide future programs to reduce perinatal outcomes for other resource-limited countries like Nepal.

## Supporting information

**S1 Data. Master data file.**
(XLSX)

## Acknowledgments

The authors are grateful to all the participating institutions and service providers who participated in the SSN newborn resuscitation and skill retention program in Madhesh province.

## Author Contributions

**Conceptualization:** Michael K. Visick, Robert B. Clark.

**Data curation:** Mala Chalise, Ranjan Dhungana.

**Formal analysis:** Mala Chalise.

**Funding acquisition:** Robert B. Clark.

**Investigation:** Ranjan Dhungana.

**Methodology:** Robert B. Clark.

**Project administration:** Ranjan Dhungana, Michael K. Visick, Robert B. Clark.

**Resources:** Robert B. Clark.

**Supervision:** Robert B. Clark.

**Visualization:** Robert B. Clark.

**Writing – original draft:** Mala Chalise, Robert B. Clark.

**Writing – review & editing:** Mala Chalise, Ranjan Dhungana, Michael K. Visick, Robert B. Clark.

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
