## [Decision Letter · Decision Letter 0]

9 Aug 2022

PGPH-D-22-00883

Assessing the effectiveness of newborn resuscitation training and skill retention program on neonatal outcomes in Madhesh Province, Nepal

Dear Dr. Robert,

Thank you for submitting your manuscript to PLOS Global Public Health. After careful consideration, I feel that it has merit but does not fully meet PLOS Global Public Health’s publication criteria as it currently stands. Therefore, I invite you to submit a revised version of the manuscript that addresses the points raised during the review process.

We look forward to receiving your revised manuscript.

Kind regards,

Rakesh Singh

Academic Editor

Journal Requirements:

1. Please update your online Competing Interests statement. If you have no competing interests to declare, please state: “The authors have declared that no competing interests exist.”

2. Please provide a complete Data Availability Statement in the submission form. If your research concerns only data provided within your submission, please write “All data are in the manuscript and supporting information files.” as your Data Availability Statement.

Review Comments:

Reviewer #1: I am glad to review this paper entitled “Assessing the effectiveness of newborn resuscitation training and skill retention program on neonatal outcomes in Madhesh Province, Nepal.”I would like to congratulate the team for doing this project to improve the quality of care in resource limited settings and identifying the area where neonatal resuscitation programme was actually required. Decreasing neonatal mortality is really a great challenge in resource limiting settings. This research has utilized evidenced based strategies and implemented a cost-effective strategies/programmer to overall improve the neonatal outcomes in this province. This paper is well written and results are well expressed.

Though this study was not a clinical study, some queries-

1. Is there data regarding baseline characteristics of neonates in baseline period and endline period? As we know, there are various factors that can affect outcome of neonates within 24 hours and after 24 hours like maternal factors and fetal factors.

2. It’s better to present the data in Table 4 as both number values and statistically significant observation for better understanding of results.

3. Minor queries: Abstract section line no 35, please mention the full form of LDS Charities here as well.

Reviewer #2: Yes , the manuscript is well written , clear and correct in most of its aspect and unambiguous. However, there were few grammatical errors which needs to be corrected and few numerical errors along with few minor changes , which I have attached as an attachment .

---

## [Decision Letter · Decision Letter 1]

23 Sep 2022

Assessing the effectiveness of newborn resuscitation training and skill retention program on neonatal outcomes in Madhesh Province, Nepal

PGPH-D-22-00883R1

Dear Dr. Clark,

We are pleased to inform you that your manuscript 'Assessing the effectiveness of newborn resuscitation training and skill retention program on neonatal outcomes in Madhesh Province, Nepal' has been provisionally accepted for publication in PLOS Global Public Health.

Best regards,

Rakesh Singh

Academic Editor

Reviewer Comments:

Reviewer #1: All comments have been addressed

Reviewer #3: All comments have been addressed